# Exploring the Feasibility of Circulating miRNAs as Diagnostic and Prognostic Biomarkers in Osteoarthritis: Challenges and Opportunities

**DOI:** 10.3390/ijms241713144

**Published:** 2023-08-24

**Authors:** Kyriacos Felekkis, Myrtani Pieri, Christos Papaneophytou

**Affiliations:** Department of Life Sciences, School of Life and Health Sciences, University of Nicosia, 46 Makedonitissas Avenue, Nicosia 2417, Cyprus; felekkis.k@unic.ac.cy (K.F.); pieri.m@unic.ac.cy (M.P.)

**Keywords:** osteoarthritis, circulating miRNAs, diagnosis, prevention

## Abstract

Osteoarthritis (OA) is a prevalent degenerative joint disease characterized by progressive cartilage degradation and joint inflammation. As the most common aging-related joint disease, OA is marked by inadequate extracellular matrix synthesis and the breakdown of articular cartilage. However, traditional diagnostic methods for OA, relying on clinical assessments and radiographic imaging, often need to catch up in detecting early-stage disease or i accurately predicting its progression. Consequently, there is a growing interest in identifying reliable biomarkers that can facilitate early diagnosis and prognosis of OA. MicroRNAs (miRNAs) have emerged as potential candidates due to their involvement in various cellular processes, including cartilage homeostasis and inflammation. This review explores the feasibility of circulating miRNAs as diagnostic and prognostic biomarkers in OA, focusing on knee OA while shedding light on the challenges and opportunities associated with their implementation in clinical practice.

## 1. Introduction

Osteoarthritis (OA) is a degenerative joint disease and the leading cause of disability in adults, causing significant pain, functional limitations, and impairments in quality of life [1]. While OA can affect nearly any joint in the body, it commonly affects the hands, knees, hips, and feet [2]. The pathological features of OA involve various joint structures, including cartilage, bone, synovium, ligament, muscle, and periarticular fat, resulting in joint dysfunction, pain, stiffness, functional limitations, and loss of mobility [3]. Primary clinical symptoms include chronic pain, joint instability, stiffness, joint deformities, and radiographic joint space narrowing [4].

The severity of OA is evaluated through clinical examination of symptoms like pain, joint stiffness, and restrictions in normal functioning, as well as through imaging tests such as X-rays or magnetic resonance images (MRIs), using the Kellgren and Lawrence (KL) grading system [5]. The KL scale ranges from 0 to 4, where 0 represents a healthy joint with no OA signs, and grades 1 through 4 show progressively worsening disease severity, with a score above 2 indicating the presence of OA. However, it is essential to note that radiographic imaging mainly captures late-stage changes in bone and cartilage [6].

The activity of chondrocytes regulates the balance between anabolic and catabolic processes in articular cartilage homeostasis. Altered joint loading due to malalignment or trauma can tip this balance towards catabolic activities, leading to joint abnormalities such as synovitis, cartilage degradation, subchondral bone sclerosis, and osteophyte formation, collectively referred to as OA [7,8]. The etiology of OA is complex and multifactorial, involving age, sex, obesity, joint trauma, environmental factors, and genetic factors, with the latter being one of the best-known contributors to OA pathogenesis [9].

Existing OA therapies aim to alleviate symptoms like pain, offering only modest, long-term benefits and leaving many patients dissatisfied with their effectiveness [10]. As mentioned above, the diagnosis of OA relies on radiographic criteria and clinical symptoms [11], with radiography remaining the primary imaging modality for diagnosis and followup of OA [12]. However, limitations of radiography have sparked interest in alternate parameters for monitoring OA, potentially serving as biomarkers in drug development [13].

Biochemical markers in blood, urine, or synovial fluid (SF) samples reflecting dynamic changes in joint remodeling and disease progression have emerged as alternatives [14]. In OA, a biochemical marker may function as a molecule, directly causing joint damage, a byproduct of such damage, or both [15]. For instance, fragments of cartilage extracellular matrix like hyaluronan can act in dual roles: they indicate the disease and instigate the ongoing immune-mediated wound-healing response in joints affected by osteoarthritis [16].

Recent studies also underscore the regulatory roles of non-coding RNAs (ncRNAs) in OA development [17]. Different classes of ncRNAs, including long ncRNAs (lncRNAs), microRNAs (miRNAs or miRs), and circular RNAs (circRNAs), have been implicated in the development of OA [17]. Specifically, these ncRNAs can influence the expression of genes that regulate inflammation, cell death, and matrix degradation in chondrocytes, thereby impacting the progression of OA [17,18].

Furthermore, circulating ncRNAs (in biological fluids/plasma/serum) exhibit high potential as biomarkers for OA due to the easy accessibility of the sample [19]. Interestingly, the primary roles of the lncRNAs and circRNAs are to function as molecular absorbers for miRNAs. Together, these ncRNAs impact the proliferation and apoptosis of chondrocytes, inflammatory reactions, and the breakdown of the extracellular matrix [17]. Nevertheless, miRNAs in the circulation (circulating miRNAs/c-miRNAs) have attracted particular attention due to their unique stability among the different classes of ncRNA [20].

Despite the identification of several genetic (e.g., *COL6A3* and *ACTG1* genes, DNA methylation, etc.) [21,22] and biochemical biomarkers (e.g., proinflammatory cytokines, C-telopeptide fragments of type II collagen, hyaluronan, etc.) [23,24], miRNAs offer a unique advantage as biomarkers for OA. Compared to genetic markers, which are “static”, miRNAs provide a dynamic and real-time view of disease states, reflecting the complex and ever-changing regulatory networks within the disease pathology [25]. Unlike protein biomarkers, which often struggle to find accurate detection methods due to low concentrations in several cases [26], miRNAs can be more easily detected and quantified using advanced techniques such as RT-qPCR and next-generation sequencing (NGS) [27]. MiRNAs also offer the advantage of potentially serving as components in multimarker models, enhancing precision in diagnosis, treatment guidance, and assessment of treatment response. In contrast to analyzing multiple protein markers, which can be costly and time-intensive, employing panels that include a variety of miRNAs may furnish a more cost-effective and non-invasive approach for diagnosing and forecasting the progression of the disease [28]. Notably, the stability of miRNAs in various body fluids enables non-invasive monitoring of several diseases, including OA, while their high sensitivity and specificity to particular pathological conditions present opportunities for more targeted diagnosis and therapy [28].

This review explores the current status, future perspectives, and limitations of c- miRNAs as prognostic and diagnostic biomarkers for OA, focusing on knee OA (KOA). We searched Pubmed, Scopus, and Science Direct databases for publications up to May 2023. “Osteoarthritis”, “non-coding RNAs”, “circulating non-coding RNAs”, “micro-RNAs”, “circulating miRNAs”, “biomarkers”, “prognosis”, “diagnosis”, and their respective medical subject headings (MeSH) terms were used as keywords. We also selected studies on c-miRNAs linked to OA. Only studies (both basic and clinical) published in English were included in this review.

## 2. MicroRNAs: Basic Concepts

### 2.1. MicroRNA Biogenesis and Biological Roles

MiRNAs are small non-coding RNA strands, approximately 20–25 nucleotides in length, that act as key post-transcriptional regulators of gene expression. They function by binding to the 3’-untranslated regions of target mRNAs, leading to their degradation or inhibition of translation [29,30]. They originate in the nucleus, transcribed by either RNA polymerase II or III into a lengthy precursor molecule called primary miRNA (pri-miRNA) [31]. This pri-miRNA undergoes processing by the Drosha-DGCR8 complex, an RNase III-type enzyme, leading to the formation of a roughly 70–100 nucleotide precursor known as pre-miRNA [29,32,33]. This pre-miRNA is subsequently exported into the cytoplasm through exportin 5 [34,35]. The ribonuclease Dicer processes pre-miRNA in the cytoplasm, resulting in a miRNA duplex of about 22 nucleotides [36,37]. One strand from this duplex is then integrated into the RNA-induced silencing complex (RISC), which is guided to target mRNA molecules [38,39,40]. MiRNAs play a crucial role in the post-transcriptional regulation of gene expression, targeting nearly 30% of the protein-coding genes by creating a complementary base pair with the 3’-untranslated region (UTR) of the target mRNA. This interaction results in translational suppression or direct mRNA degradation [41]. So far, over 1600 human miRNAs have been identified (reviewed in [30]), and their significant contributions to a diverse range of developmental processes and pathologies have been widely recognized [42]. For example, miRNA-21 has been implicated in the development of both neoplastic and non-neoplastic diseases. Specifically, the downregulation of miR-21 has been observed to increase the rate of cell death by targeting genes involved in inflammation and cancer, such as HIF-1α, PTEN, and PDCD4 [43,44]. Conversely, miR-21 has also increased cell migration by targeting TPM1 and PDCD4 in neoplastic diseases [45]. These findings highlight the multifaceted roles that miR-21 plays in various pathological conditions, demonstrating its potential as both a diagnostic and therapeutic target. Additionally, miR-206 has been shown to affect muscle cell function significantly. Specifically, the overexpression of miR-206 suppressed muscle cell proliferation and induced cell cycle arrest in the G0/G1 phase. This effect was mediated by inhibition of the glucose-6-phosphate dehydrogenase (G6PD) gene, highlighting the precise regulatory role of miR-206 in cellular growth and development [46]. In addition to skeletal muscle development, miR-206 acts as a tumor suppressor by regulating the expression of different genes, such as *Sox9* [47], *Cyclin D2* [48], *Notch 3* [49], and *FMNL2* [50], in various tissues. miRNA-206 is also downregulated and suppresses cell proliferation by targeting Forkhead box protein 1 (FOXP1) in brain gliomas [51].

Some miRNAs synthesized within cells can be actively secreted into the bloodstream, thereby increasing the concentration of these corresponding miRNAs. As previously mentioned, these miRNAs are known as circulating miRNAs (c-miRNAs). These c-miRNAs can then reach receptor cells and perform distal regulatory functions. Present in virtually all biological fluids, c-miRNAs are gaining recognition as potential non-invasive biomarkers for numerous human pathologies, including cancer, cardiovascular diseases, and infectious diseases [52]. Although most miRNAs are located intracellularly, these c-miRNAs found in plasma, serum, and other body fluids demonstrate stable existence despite the presence of RNases [52]. Remarkably, they withstand even severe conditions, such as repeated boiling, extreme pH values, freezing–thawing cycles, and room temperature storage, ensuring their reliability [53].

Intriguingly, c-miRNAs are primarily found within exosomes, accounting for about 83–99% of the total miRNA population detected in circulation [54]. These exosomal miRNAs include AGO2-bound miRNAs, free miRNAs, and mature miRISCs [55]. Exosomes are extracellular vesicles (Evs) (ranging from 40 to 100 nm in diameter) that play an essential role in the transport and stability of miRNAs in biological fluids, including blood, urine, and saliva [56,57]. These specific EVs are lipid bilayer-enclosed structures actively secreted by cells into the extracellular environment. Packaging various molecules, including proteins lipids, messenger RNAs, and ncRNAs, in exosomes protects them from enzymatic degradation [58]. Furthermore, it facilitates their transport to recipient cells. Due to specific proteins on the surface of exosomes, these EVs engage with target cells through a ligand–receptor interaction, leading to endocytosis. This process enables the delivery of encapsulated materials from one cell to another [59]. Exosomes play a role as mediators of cell-to-cell communication, exhibiting pleiotropic activities and participating in homeostasis regulation. Exosomal miRNAs involve various physiological and pathological processes, including inflammation, tissue repair, immune response, and cancer progression (for a review on the topic, see Ref. [60]). Notably, miRNAs are the cargo of exosomes that have attracted specific attention [61]. Encapsulation of miRNAs within exosomes enables precise targeting and cell-to-cell communication [62]. In musculoskeletal diseases, including osteoarthritis and osteoporosis, exosomes derived from host cells and gut microbiota have been explored as potential nanocarriers for targeted therapy [63].

Another key factor contributing to their stability is that both intravesicular and extravesicular c-miRNAs are frequently protein-bound. Research has indicated that about 90% of c-miRNAs are associated with RNA-binding proteins (RBPs), such as the AGO2 protein [64], high-density lipoprotein (HDL) [65], and low-density lipoprotein (LDL) [66], with most circulating miRNAs being transported as AGO2-bound miRNAs [64].

The dysregulation of miRNAs in various diseases, including OA, signifies their potential as diagnostic and prognostic biomarkers [20,67]. Regarding OA, altered expression of miRNAs can influence key pathological processes like cartilage degradation and inflammation, making them strong candidates for non-invasive biomarkers [68]. In addition to their diverse roles in regulating biological processes, miRNAs have pivotal roles in OA progression, and abnormal expressions of specific miRNAs can reflect pathological changes in OA [69] (discussed further below).

### 2.2. The Roles of miRNAs in Bone Development

Bone development begins with the establishment of mesenchymal stem cell (MSC) condensations that prefigure the shape, size, and location of mature bone elements [9]. Recent studies have elucidated that miRNAs participate in different stages of bone growth and development [10,11], and reviews have highlighted their role in regulating bone formation [70,71]. Several in vivo studies using animal models have shed light on the role of miRNAs in bone formation. For example, the expression of members of the miR-34 family (miRs-34a, b, and c) was found to increase during osteoblast differentiation of murine calvarial cells [72]. These miRNAs inhibited osteoblast proliferation by suppressing cyclin D1, CDK4, and CDK6 accumulation, as well as terminal differentiation, at least partly through the inhibition of SATB2, a nuclear matrix protein critical for osteoblast differentiation [72]. Deletion of miR-34b and miR-34c in Col1a1-producing cells in mice led to increased bone mass during embryonic development and postnatally, while the opposite was observed when miR-34c was overexpressed [72]. The essential role of miR-206 in bone development, particularly in osteoblast differentiation, was highlighted by Inose and coworkers [73]. They demonstrated that the expression of miR-206 in osteoblasts decreased over the course of differentiation and that overexpression inhibited differentiation while knockdown promoted it.

Additionally, the deletion of the miR-181 family in mice resulted in smaller sizes among survivors [74]. Another study revealed that miR-182 is a crucial osteoclastogenic regulator in bone homeostasis, inhibiting osteoclastogenesis in vitro and affecting trabecular bone mass in vivo [75]. Furthermore, miR-21 appeared to enhance both osteogenesis and osteoclastogenesis [76,77]. Its global knockout in mice did not appear to affect bone development. Still, it did promote trabecular bone mass with age and prevented OVX-induced bone loss during aging, partially due to suppressed osteoclast function [78]. Crosstalk mediated by miRNAs in bone and exosomes containing miRNAs derived from differentiated bone cells can regulate other cell types involved in bone formation and turnover [21,22]. For example, miR-23a-5p from osteoclast-derived exosomes can suppress osteoblast differentiation, in part, by targeting Runx2 [79].

### 2.3. The Roles of miRNAs in OA Pathogenesis

The vital role of miRNAs in the development of the musculoskeletal system and their contribution to OA pathology was first brought to light through genetically modified animal models [80]. These models underwent either a knockout or overexpression of key genes involved in miRNA biogenesis and processing. Focusing specifically on chondrocyte homeostasis, studies using tissue-specific knockout mice have underscored the significance of miRNA processing enzymes [80]. A particular example is Dicer, a crucial enzyme for miRNA biogenesis; its deficiency in Col2α1-expressing cells leads to early postnatal fatality and aberrant skeletal growth due to decreased proliferating chondrocytes [81]. In detail, these results were obtained using a Dicer-null mouse model, which displayed diminished proliferating chondrocytes in its growth plates and hastened differentiation into a hypertrophic type. These alterations led to significant skeletal growth anomalies and premature death, thereby illuminating the critical role of miRNA in chondrocytes and, by extension, in OA [81]. Moreover, miRNAs have a crucial function in cartilage formation and, consequently, significantly influence the pathophysiology of OA. Specifically, variations in the expression levels of specific miRNAs can disrupt the balance in chondrocyte function by impacting apoptosis, senescence, and autophagy and can enhance the level of inflammation and cartilage matrix breakdown by promoting the production of proinflammatory agents (like TNF-α, IL-1, IL-6, COX-2, NO, and ROS) and matrix metalloproteinases (MMPs) [82].

Furthermore, it has been demonstrated that some miRNAs can regulate the expression of inflammation factors in OA, such as miR-146a [83], miR-142-3p [84], and miR-130a [85]. Notably, the expression of miR-140 is unique to cartilage; however, its function in growth and/or tissue preservation is still uncertain [86]. In contrast, miR-146a displays increased expression levels in OA cartilage compared to healthy cartilage [83]. Interestingly, miR-146a-5p was found to be significantly overexpressed in OA patients both in the articular cartilage tissue and in serum, with a positive correlation observed between levels in both sample types. These data imply that serum levels of miR-146a-5p can mirror the molecular activities in the cartilage, suggesting its potential use as a clinical biomarker for the management of OA [87]. Furthermore, miR-146a has garnered significant interest due to the consistency of reports of its deregulation in OA across multiple studies. For instance, in the case of rheumatoid arthritis (RA), miR-146 was observed to be overexpressed in synovial tissue following stimulation with inflammatory cytokines tumor necrosis factor-alpha (TNF-α) and interleukin-1β (IL-1β) [88]. It was previously demonstrated that miR-146a targets TRAF6 and IRAK1, playing a significant role in inflammation control [89].

Song et al. [90] demonstrated that miR-21 was suppressed in OA patients and that the modulation of miR-21 influenced apoptosis and autophagy of OA chondrocytes. Interestingly, growth-arrest-specific 5 (GAS5), a long ncRNA, plays a pivotal role in controlling miR-21 during the progression of osteoarthritis [90]. It has been suggested that GAS5 contributes to the pathogenesis of OA by acting as a sponge for miR-21, thereby influencing cell survival. Nevertheless, the potential interaction network between miR-21 and GAS5 remains inconclusive. It is worth mentioning that miR-21 has drawn attention due to its significant role in developing and spreading tumors, particularly in cell proliferation and differentiation processes essential for chondrogenesis and cartilage remodeling [91]. Despite this, the specific role and underlying mechanism of miR-21 in OA progression remain uncertain. In addition, GDF-5 functions as a signaling molecule for chondrogenesis, facilitating chondrocyte differentiation [92].

Moreover, several recent reviews have highlighted the contribution of miRNAs in normal conditions of the articular cartilage (AC) (osteoclastogenesis, osteoblastogenesis, and chondrogenesis), as well as in pathological conditions (cartilage degradation, synovial inflammation, and OA progression) [93,94]. Numerous miRNAs, including miR-101 [95], miR-181c [96], miR-675 [97], miR-770 [98], miR-140 [86], miR-146a [99], miR-27a [100], miR-122 [101], miR-130a [85], miR-15a [102], and miR-127-5P [103] have been reported to be engaged in OA pathogenesis through regulation of cartilage homeostasis, chondrocyte metabolism, and inflammatory responses, as well as proteolytic enzyme activity. A recent meta-analysis of 191 miRNAs by Liu and coworkers [104] revealed that several miRNAs consistently exhibited altered expression in OA. Remarkably, miR-146a-5p, miR-34a-5p, miR-127-5p, and miR-140-5p emerged as prominent candidates with potential diagnostic and prognostic value. The study also highlighted the significance of downstream effectors, including mesenchymal stem cells and transforming growth factor-β, regulated by miRNAs in OA progression. These findings underscore the importance of miRNA signaling and provide insights into potential biomarkers and therapeutic targets for OA. For example, miR-140 targets ADAMTS-5 and has been implicated in chondrogenesis, providing an essential link between miRNA regulation and cartilage homeostasis [105]. Other recent reviews underscore the significant role of miRNAs in the onset and progression of OA. In one such study, Boehme and Ralauffs [106] emphasized the importance of 27 miRNAs or miRNA families, which exhibit significant upregulation or downregulation in human AC of OA patients compared to normal AC. The authors discussed how these differentially regulated miRNAs interfere with the downstream signaling of inflammatory cytokines, fibroblast growth factor 2 (FGF2), and transforming growth factor β (TGF-β), displaying significant interplay at the protein level. In the context of OA AC, procatabolic target gene expression is facilitated by NF-κB, MAPK, and SMAD signaling, while proinflammatory targets are activated downstream of MAPK and NF-κ B signaling. Interestingly, although cell proliferation is dominant during the early stages of OA, catabolic proteins are simultaneously induced, which degrade the extracellular matrix (ECM).

The essential role of FGF signaling in correctly forming and maintaining articular cartilage is evident, as irregular FGF signaling can lead to joint malformation and OA initiation and advancement [107]. Research involving FGF2 transduction in human knee articular cartilage samples has illustrated how it can emulate the early stages of OA by triggering cell proliferation, influenced by miRNAs such as miR-140, which has been shown to regulate Wnt signaling, often interconnected with FGF signaling [86,108]. Additionally, the TGF-β superfamily, comprising 42 human growth factors and categorized into two distinct subfamilies, has garnered considerable interest due to its multifaceted roles in the physiological conditions of joints and the onset of OA [109]. The subfamilies are the TGF-β/activin/nodal and the BMP/GDF/MIS branches [110]. The impact of TGF-β on numerous cellular functions, including cell growth, is paramount, and any disruptions in its signaling pathways could have substantial repercussions for OA [111]. MiR-146a has been implicated in modulating TGF-β signaling in OA by targeting SMAD4 [112], and miR-21 has been shown to influence MSC osteogenic differentiation through the TGF-β signaling pathways [113]. Furthermore, bone regeneration relies on the activation and transformation of MSCs, which fulfill essential immunomodulatory and regenerative functions by evolving into osteoblastic cells that form and mineralize the bone matrix [114]. The role of miRNAs in guiding the development process, including the function of MSCs in bone formation, has been emphasized in recent research, with miR-21 playing a significant role in MSC differentiation [115].

Table 1 shows some examples of miRNAs that have been linked to the development of OA. These miRNAs can be attributed to tissue damage due to direct cartilage injury, persistent inflammation, apoptosis, or even cells affected during the OA process that could potentially release these miRNAs [20]. These miRNAs have been identified in tissues and/or biological fluids using different methods and protocols.

## 3. The Potential of Circulating miRNA as Biomarkers for Osteoarthritis

As previously mentioned, some miRNAs involved in the pathogenesis of OA can be detected in the blood or other biological fluids. These c-miRNAs can originate from various sources, including tissue damage from direct cartilage lesions, chronic inflammation, and apoptosis, or cells affected by the OA process that release these miRNAs. Thus c-miRNAs can be discharged into the extracellular space surrounding the chondrocyte, with the specific type of miRNA potentially being influenced by the stage of the disease [118]. Recent studies have explored the potential of c-miRNAs as diagnostic markers in OA and analyzed their expression levels in the circulation and bone/cartilage to understand the biological process occurring in different pathologies. Several studies aimed at validating c-miRNAs in plasma or serum, as well as whole blood, as prognostic or diagnostic markers for the discrimination of OA patients from healthy individuals [68]. Over the years, additional research has reinforced the significance of miRNA in the onset and progression of OA.

Jones et al. [116] determined a set of 17 miRNAs that contribute to cartilage alteration by profiling 157 miRNAs; the authors suggested that the altered expression of these miRNA could be a consequence of hypomethylated states in the miRNA promoter regions. Some specific miRNAs have also shown essential roles in OA; for example, downregulation of miR-140 inhibits interleukin-1β (IL-1β) by inducing *ADAMTS* expression, miR-27b regulates *MMP-13* expression in human chondrocytes and is downregulated in patients with OA, and miR-146a is overexpressed in patients with low-grade OA, suggesting its involvement in OA pathogenesis [86,130,131].

Furthermore, miRNAs found in SF, such as miR-29b3p and miR-140, strongly correlated with radiographic KOA severity and were therefore suggested as potential OA biomarkers [132]. Castanheira and coworkers [133] reported a significant reduction in miR-223 in SFs obtained from patients with early OA, while miR-23b, let-7a-2, snord96A, and snord13 were significantly increased. Notably, it has been suggested that let-7a controls the IL-6 receptor (IL6R). Its suppression can enhance cell growth, reduce cell death, and inhibit inflammation in an in vitro OA model [134]. The let-7 family is frequently mentioned in OA research; a broad study found that serum let-7e can potentially forecast OA risk independent of age, gender, and body mass index [135]. A recent study also affirmed this, noting diminished let-7e expression in the serum of KOA patients [135]. Although the exact functions of let-7 family miRNAs are still unknown, their potential as OA biomarkers is increasingly recognized. Similarly, OA-specific miRNAs were identified in SF that are differentially regulated in early- and late-phase OA [124,125]. Ntoumou et al. [136] found 279 miRNAs differentially expressed in the serum of osteoarthritic conditions, and three signature miRNAs (140-3p, 33b-3p, and 671-3p) were downregulated. These miRNAs are known to be involved in several molecular pathways, including Wnt, ErbB, and TGF-beta.

However, the utilization of c-miRNAs (found in blood/plasma/serum) as OA biomarkers stems from the simplicity of sample acquisition. While SF samples have been utilized for similar purposes, collecting blood/plasma/serum samples has a clear advantage due to their minimally invasive extraction procedures [137]. In detail, Murata et al. [120] demonstrated that differentially expressed miRNAs, especially miR-132, in the plasma and SF can distinguish patients with OA or RA from healthy controls. Borgonio- Cuarda and coworkers [118] identified ten circulating miRNAs (miR-16, -29c, -93, -126, -146a, -184, -186, -195, -345, and -885-5p) in the plasma of patients with primary OA. Notably, the potential of miR-16 and miR-146a as biomarkers for OA has been highlighted by other authors [120,131]. Similar to Murata et al. [120], in their study, Borgonio-Cuadra et al. [118] also observed the deregulation of miR-16 and miR-146. Interestingly, miRNAs were detected in both plasma and SF samples; however, in plasma, the levels were higher than in SF [120]. It has also been demonstrated that knee and hip AC of OA patients exhibit elevated miR-16 expression compared to healthy AC [119]. SMAD3 is a direct target of miR-16, implicating this miRNA in the switch to catabolic TGF signaling [121]. Notably, FGF2 is a direct transcriptional target of miR-16 in human nasopharyngeal carcinoma cells [138], indicating the additional repression of FGF2 signaling by this miRNA. In addition, Iliopoulos et al. [119] reported increased expression of miR-16 in chondrocytes of OA patients with KL scores of 3 and 4. In the same study, the expression of 365 miRNA in articular cartilage tissues from patients with OA was examined, and 16 miRNAs were altered, representing one of the first miRNA signatures to distinguish osteoarthritic from normal chondrocytes [119]. Notably, it has been demonstrated that miR-146b regulates cell differentiation, proliferation, and migration in various tumor cells [139,140]. MiR-146b has also been implicated in the chondrogenic differentiation of human bone marrow-derived skeletal stem cells (SSCs) through the modulation of SOX5 [141].

In another study, Wan et al. [142] reported a significant reduction in plasma miR-136 in KOA patients, correlating negatively with the severity of the disease (KL score). Notably, the diminished levels of plasma miR-136 exhibited a significant capability to distinguish OA patients of KL grades 2, 3, and 4 from healthy individuals. Concurrently, serum interleukin (IL)-17 levels, a target gene of miR-136, increased with KOA severity. Together, the above data suggest that decreased plasma miR-136 levels could serve as a potential biomarker for KOA. On the contrary, Beyer et al. [143] demonstrated that the levels of serum let-7e have an inverse correlation with OA severity and the number of knee/hip joint replacements. In their study, Kong et al. [144] reported that the circulating levels of miR-486-5p, miR-320b, miR-122-5p, miR-92a-3p, and miR-19b-3p were elevated in OA patients compared to the control group. A combination of miR-486-5p, miR-122-5p, and miR-19b-3p provided the best diagnostic efficacy, demonstrating a significant correlation with the risk and severity of KOA. Likewise, a study by Ntomou et al. [136] revealed that decreased miR-671-3p, miR-140-3p, and miR-33b-3p serum levels can be used to identify OA patients. These reduced miRNA levels were correlated with the risk and progression of OA.

Interestingly, in a cohort of postmenopausal women, elevated levels of circulating miR-146a-5p and miR-186-5p were associated with existing KOA and the onset of KOA over the following four years, respectively [128], suggesting that circulating miR-136 and the combination of miR-486-5p, miR-122-5p, and miR-19b-3 hold the most significant potential for distinguishing OA patients from healthy individuals [142,144]. Among the validated miRNAs mentioned earlier, miR-146a levels were elevated in the plasma of OA patients compared to controls [118] and in the serum of individuals with prevalent KOA versus healthy controls [128]. The sensitivity and specificity of the most promising circulating miRNAs that can be used as biomarkers for OA were recently thoroughly reviewed by Bottani et al. [68] and are not discussed here.

### 3.1. Challenges in Using c-miRNAs as Biomarkers for OA: Why Are c-miRNAs Not Being Used in Clinical Applications Yet?

Despite the promising prospect of c-miRNAs serving as prognostic and diagnostic biomarkers for OA and various other bone disorders, these c-miRNAs have not been incorporated into clinical practice to date. Figure 1 presents a SWOT analysis, demonstrating the potential viability of c-miRNAs as biomarkers for monitoring the onset and progression of OA.

#### 3.1.1. Strengths

To achieve the objectives of personalized medicine, new and more accurate biomarkers for the prognosis and diagnosis of OA are sorely needed. An ideal biomarker must meet specific prerequisites. First, it should be easily accessible, which means it can be identified and measured through minimally invasive procedures. The biomarker’s specificity to the disease being investigated is another vital factor, coupled with its sensitivity (ideally, it should be detectable even before the clinical symptoms have surfaced, and its quantities should be consistent with the progression of the disease or response to treatment). Lastly, the biomarker should potentially translate from research to clinical practice [145].

As extensively discussed herein, c-miRNAs meet the above criteria, as they are found in circulation, including whole blood, serum, plasma, and other biological fluids, and exhibit stability under extreme conditions [146]. Identifying miRNAs in biological fluids relies on RT-qPCR, which offers highly reproducible outcomes and enhanced specificity and sensitivity [147]. Notably, a number of these c-miRNAs are tissue-specific. Moreover, others can help distinguish patients with OA from healthy individuals or patients suffering from bone-related diseases such as RA [120].

#### 3.1.2. Weaknesses

However, in this study, we have pinpointed several differences and challenges within the research aimed at identifying and validating circulating miRNAs as prognostic and diagnostic biomarkers for OA. Some examples are presented in Table 2 and discussed in the following paragraphs.

We have identified significant variations in four levels: (i) study design and cohort selection, (ii) preanalytical stage, (iii) analytical stage, and (iv) postanalytical stage, including data analysis and normalization (Figure 2). These variations may lead to inconsistencies among studies aiming to examine the prognostic potential of specific miRNAs in OA. An example is illustrated in Table 1 by comparing the studies conducted by Borgonio Cuadra et al. [118] and Murata et al. [120], both examining the role of miR16 in KOA. Interestingly, while Borgonio Cuadra et al. [118] reported that c-miR-16 is downregulated in KOA patients, Murata et al. [120] reported that the same miR was upregulated. As illustrated in Table 1, the two studies used different methods to extract, quantify, and normalize miRNAs.

As discussed in the following paragraphs, the lack of standardized protocols and guidelines has significantly contributed to the delay in transitioning c-miRNAs from basic research to clinical application.

(i)Study design and cohort selection: For clinical application, the most critical evaluation criteria for c-miRNAs as diagnostic and prognostic biomarkers are high sensitivity and specificity to avoid false-positive or false-negative diagnoses. To this end, it is essential to have a larger sample size to be able to discriminate OA patients from healthy controls or assess the severity of the disease [28]. Given the numerous criteria used in clinical applications, such as age, gender, ethnicity, lifestyle, and medical history, it is essential to avoid studies with limited sample sizes. The severity of OA is another factor that should be taken into account. As mentioned in the Introduction, the severity of KOA is assessed using a scale that categorizes the severity of KOA into five grades, ranging from grade 0 (normal) to grade 4 (severe) through visual inspection of X-ray or MRI images [5].

Examples of differences in cohort selection among studies are shown in Table 2. For example, the number of participants in some studies was low, with only male or female participants. In addition, some studies did not report the severity of OA (e.g., KL score). As also shown in Table 2, one study involved participants who received a specific OA drug. Notably, one study did not report the type of OA (i.e., knee, hip, hand, etc.) or the severity of the disease.

Another factor that must be considered is the significant role inflammation plays in OA development, particularly during the disease’s early stages (early OA) [149]. Previous studies have shown that inflammatory processes during early stages can contribute to cartilage degradation and changes in the subchondral bone [150,151,152]. Consequently, the severity of the disease (e.g., as indicated by the KL grade) should be considered when recruiting participants. It is essential to note that the inflammation status in OA can differ based on the severity of the disease. Recently, we proposed that the observed variability in exosomal miRNA levels in patient samples compared to healthy individuals might be partly due to interactions between exosomes carrying c-miRNAs and endothelial receptors. These receptors are often overexpressed under pathological conditions such as inflammation [153].

The participant’s physical activity (PA) level is another crucial factor affecting miRNAs concentration and profile in the blood. Whether regularly exercising or sedentary, a person’s PA state significantly affects the expression of miRNAs across various tissues and organs. Consequently, it also impacts the profile of c-miRNAs [154]. Numerous miRNA alterations in human skeletal muscle and circulation have been recorded in response to acute and chronic exercise [155]. Studies have shown changes in c-miRNA levels post-exercise, such as the downregulation of miR-16 and miR-93 and a decrease in miR-222. Notably, miR-16 targets the vascular endothelial growth factor/vascular endothelial growth factor receptor pathway, which plays a role in adapting to different types of physical activity. In contrast, miR-93 modulates mitogen- and stress-activated protein kinase 2 (Msk2), a histone kinase activated by muscle contraction [156]. Interestingly, specific miRNAs required for skeletal muscle function, including miR-23a,-133a, -146a, -206, -78b, and -486, showed significant changes within the muscle after a single bout of acute resistance exercise. However, these miRNAs did not show apparent alterations in the plasma post exercise. This observation suggests that c-miRNA levels may not accurately reflect the miRNA responses within skeletal muscle after exercise [157].

Finally, specific drugs such as heparin, aspirin, and antiplatelet therapies may affect the profile and/or concentration of miRNAs [158,159]. In detail, heparin, an anticoagulant frequently used in myocardial infarction treatments, is known to inhibit qPCR-based ncRNA quantification [160]. Administering heparin before blood collection has also been demonstrated to affect the levels of miRNAs, while endogenous heparin also impacts miRNA quantification [161]. Aspirin and antiplatelet treatments also influence miRNA profiles. For example, aspirin alters the expression of platelet-derived miRNAs, such as miR-19b and miR-92a, as well as others, including miR-155, miR-21, miR-98, miR-191, miR-126, miR-223, and miR-150 [159,162]. Other antiplatelet therapies impact the levels of some miRNAs, including miR-191, miR-223, miR-126, and miR-150 [160]. Consequently, special consideration must be taken during patient recruitment for miRNA quantification studies, especially regarding the administration of heparin and antiplatelet therapies before blood collection. It is noteworthy that treating blood samples with the enzyme heparinase can reverse heparin’s effects [140]. This more profound insight into the interaction between these drugs and miRNAs emphasizes the complexity of their role in diagnostic and therapeutic applications [163].

(ii)Preanalytical stage: Several preanalytical factors may affect the profiles and levels of c-miRNAs. We recently reviewed the main preanalytical factors affecting the profile and concentration of miRNAs in circulation when they are examined as potential biomarkers for cardiovascular diseases (CVDs) [163], which are also summarized herein. One of the most critical factors is the selection of the blood fraction (whole blood, plasma, or serum), sample collection (e.g., needle gauge), anticoagulant (for plasma collection), centrifugation conditions, and handling/storage conditions of the samples (Figure 2). These factors significantly impact miRNA profiles and are usually overlooked. Furthermore, based on the results of this work and our previous work [163] and that of others [68], the sample source is one of the most critical aspects in determining c-miRNA concentrations [164] for various diseases, including OA. Plasma is often favored over serum as a source of c-mRNAs, since the coagulation process can release RNA molecules, potentially altering the genuine profile of c-miRNAs. However, plasma may contain cellular components such as apoptotic or lysed cells (e.g., red blood cells-RBCs and platelets) that may contribute miRNAs. Furthermore, anticoagulants like citrate and heparin citrate, which are used to isolate plasma, can inhibit downstream methodologies, including RT-qRCR. Therefore, serum is often deemed the optimal fraction for detecting c-miRNAs [165,166]. Furthermore, using whole blood as a source of miRNAs should be avoided because cellular fraction could also contribute miRNAs [167]. Therefore, we suggest that results from studies employing different blood fractions and types of blood tubes not be compared directly. Most crucially, only miRNAs that are not marginally up- or downregulated are likely to be suitable as clinical biomarkers. However, as presented in Table 1 and Table 2, various blood fractions have been used in different studies investigating the potential of c-miRNAs as biomarkers for OA. Furthermore, in some studies, the type of collection tube used and the centrifugation conditions were not reported (Table 1 and Table 2).

The extraction method/protocol is another factor affecting the yield and profile of c-miRNAs. As illustrated in Table 1 and Table 2, the extraction of miRNAs/c-miRNAs in the studies was carried out with either TRIzol-based methods or commercial-column-based extraction kits.

Despite efforts to compare and optimize these methods, interlaboratory differences persist across isolation methods, yielding varied miRNA quantities and qualities (reviewed in [153]). For example, studies have found that TRIzol-based methods negatively impact RNA quality and that column-based methods are preferred. Similarly, two column-based RNA extraction kits (miRNeasy Mini and mirVana PARIS) reported higher RNA yields than the TRIzol-based extraction method [168]. To avoid these discrepancies, the use of equal volumes of starting material (serum or plasma) has been suggested instead of using the same amount of total RNA to ensure accurate results [167].

Thus, standardizing both sample collection and handling protocols would be beneficial to reduce the bias that can affect the preanalytical phase of miRNA validation [169]. Different research groups have endeavored to standardize the steps involved in this stage, encompassing the collection, processing, and storage of blood samples, to mitigate potential inconsistencies and variations in c-miRNA profiles. However, a universally accepted and reliable method for the processing of blood for c-miRNA analysis has not been established. Consequently, scientific publications must thoroughly describe the procedures for blood collection and processing [170]. Recently, guidelines and preventive measures for the collection and handling of blood samples have been proposed by different groups [171,172]. The recommended steps involve (i) utilizing EDTA-treated tubes for blood collection (for plasma samples), (ii) transporting blood samples to the laboratory with minimal agitation and at low temperature, (iii) immediate centrifugation of the samples, (iv) careful removal of plasma followed by prompt aliquoting, (v) storage of plasma/serum samples at −80 °C, and (vi) using pre-aliquoted samples to avoid repetitive freeze–thaw cycles. It is crucial that blood samples are processed within a restricted time frame after collection to minimize miRNA expression contamination by lysed RBCs, platelets, and leukocytes and that plasma and serum samples are cautiously recovered, aliquoted, and frozen within 24 h. Furthermore, they must be stored at −80 °C [171]. Measures should also be taken to reduce contamination by cellular material and hemolyzed samples, such as employing an additional centrifugation step after plasma/serum recovery to eliminate intact blood cells [164]. In vivo hemolysis may be unavoidable, but in vitro hemolysis can be significantly mitigated by adhering to guidelines set forth by the Early Detection Research Network (ERDN) [173]. Lastly, it is essential to refrain from combining different specimen types, such as plasma and serum, within one study, as the uniform expression of miRNAs across these fractions is not guaranteed [163].

(iii)Analytical stage: The platform used for miRNA evaluation introduces a notable source of error in the analytical phase. Common platforms include next-generation sequencing (NGS), microarrays, and RT-qPCR, each with unique pros and cons [174,175]. As illustrated in Table 1 and Table 2, most studies aimed at validating miRNAs as biomarkers for OA have employed RT-qPCR, which is cost-effective and fast but limited by low throughput. Moreover, the efficiency of each method depends on the quality of the starting material. Notably, c-miRNA levels isolated with different protocols vary significantly, and direct comparisons should be avoided [176]. Therefore, the analytical protocols and platform must be unchanged [146].(iv)Data analysis and normalization: In the post-analytical stage, reference gene selection and normalization strategy are key challenges in miRNA quantification due to the lack of a standardized methodology. RT-qPCR data for miRNA expression can be normalized using single or multiple endogenous or exogenous reference genes or the averaged expression value of all measured miRNAs (reviewed in [68,163]). As illustrated in Table 1 and Table 2, studies aimed at validating miRNA as biomarkers for OA have used a variety of normalization strategies. Given the variables affecting miRNA quantification in the preanalytical, analytical, and post-analytical stages, it is vital to establish detailed, standardized guidelines for consistent and comparable miRNA expression data across studies and labs. Establishing universal guidelines and protocols is critical for c-miRNAs to become clinically valid biomarkers. Overall, normalization is crucial when determining c-miRNAs expression levels. Although various reference genes have been proposed, further studies are needed to identify the most reliable normalization method. This might vary depending on the miRNA release route (e.g., microparticles or protein-bound) [177]. Establishing an optimal endogenous control for each type of cardiovascular disease is essential, as specific c-miRNA expression profiles and/or levels may vary. Several studies suggest the use of multiple reference genes or a suitable combination thereof and a standard concentration of spike-in miRNAs for normalization. All samples should be simultaneously processed using identical starting volumes [178].

#### 3.1.3. Opportunities

Despite the limitations mentioned above, the potential of c-miRNAs as prognostic and diagnostic biomarkers for OA is highly recognized, especially given that the field of c-miRNA research is relatively new. Recent evidence has revealed that miRNAs contribute to the development of OA via various mechanisms [179]. As mentioned above, c-miRNAs have the potential for early detection of OA and to discriminate OA patients from healthy individuals, as well as from patients suffering from other types of bone diseases. Therefore, specific c-miRNAs could be used as indicators for earlier therapeutic intervention, potentially slowing disease progression or preventing severe damage. The early identification of at-risk patients would provide an opportunity to implement lifestyle changes or therapeutic interventions that could modify the disease course. From a personalized medicine perspective, miRNA profiles could facilitate personalized treatment strategies, enabling patient-centered care. Importantly, the unique miRNAs profile of patients can help in tailoring treatments for OA [180]. Detecting c-miRNAs would improve personalized medicine since one of the current clinical challenges is developing minimally invasive approaches. They show remarkable stability in bodily fluid and consistent expression profiles, with the potential to serve as biomarkers for changes in physiological and pathological conditions [181]. The expression signatures of c-miRNAs are emerging as novel biomarkers of numerous diseases, including CVDs, diabetes, cancer, musculoskeletal disorders, etc. [182,183].

Furthermore, the rapid advancement in technologies such as NGS and bioinformatics tools could facilitate the discovery and analysis of new c-miRNAs associated with OA. These tools can help better understand the disease mechanisms and potentially uncover new therapeutic targets and/or strategies [184,185]. Novel methodologies for the normalization of c-miRNAs concentration in biological fluids are also being developed [163]. Moreover, recent advances in artificial intelligence (AI) could facilitate the unraveling of the crucial role of miRNAs in the pathophysiology of OA. For example, in their recent study, Liu et al. [186] shed light on the role of circRNA–miRNA–mRNA regulatory networks in OA using AI tools. Finally, exploring the roles of c-miRNAs and regulating the expression or function of specific c-miRNA expression might offer a potential therapeutic strategy for the treatment of OA. For example, miRNA mimics or inhibitors could be used to modulate the expression of target genes in OA [82,187].

#### 3.1.4. Threats

Finally, we identified specific threats (Figure 1) in utilizing c-miRNAs as biomarkers for OA. Research on circulating miRNAs is still in its early stages, and the human body potentially contains thousands of different miRNAs. These can have intricate interactions with each other and other types of molecules, including other ncRNAs [188]. Maintaining high analytical standards and adopting robust study designs are prerequisites for the generation of reliable data and avoiding the pitfalls of early genetic association studies. As previously discussed, the lack of standard protocols and guidelines for miRNA extraction and quantification significantly hampers the reproducibility and accuracy of results [163]. This lack of standardized methods has led to discrepancies among studies investigating the potential of miRNAs as OA biomarkers [163]. Additionally, before the clinical implementation of miRNAs as biomarkers, they must undergo rigorous clinical trial testing and gain approval from regulatory authorities. This process can be time-consuming and costly, and there is no approval guarantee for a specific biomarker [189].

Moreover, other circulating biomolecules related to bone cartilage and bone turnover (e.g., cartilage oligomeric matrix protein and procollagen type II C-terminal propeptide) are under evaluation as OA biomarkers [190]. The concentrations of these biomarkers can be easily determined using commercially available kits, without laborious extraction steps, and normalization of results is straightforward [191]. Lastly, it is important to note that miRNA profiles may overlap among different bone diseases (e.g., osteoporosis), which can limit their specificity [192].

## 4. Conclusions

This review focused on examining the potential of c-miRNAs as prognostic and diagnostic biomarkers for OA, a global endemic and debilitating disease. Previously thought to be simply the result of “wear and tear”, OA is now understood to involve a complex interplay of local and systemic factors [193]. The development of disease knowledge and treatment innovation in OA has historically been slow, partly due to the alleged lack of valid and responsive biomarkers to evaluate efficacy, which, in turn, depends on the slow evolution of our understanding of complex joint tissue biology.

An approach for optimizing OA treatment lies in identifying effective early diagnostic biomarkers. C-miRNAs fulfill the criteria of ideal biomarkers, as they are stable in biological fluids, including plasma and serum, and can easily be detected using PCR-based techniques (Figure 3). Moreover, the role of miRNAs in OA progression is increasingly recognized, and the development of therapies that modify the disease process and alleviate symptoms is of particular interest [82]. Since OA is the primary cause of joint damage and irreversible disability, early diagnosis and aggressive treatment management of OA are necessary to improve the disease prognosis [69]. Emerging evidence has revealed that the identification of biomarkers, including c-miRNAs, could improve OA diagnosis and treatment [69].

The challenge lies in identifying specific miRNA patterns that can differentiate between various conditions, including distinguishing OA patients from healthy individuals, assessing the severity of the disease, and discriminating OA patients from those with other bone diseases. However, it is worth noting that despite these overlaps, unique patterns of miRNAs could still be identified in specific diseases when a broad enough panel of miRNAs is examined. That could potentially increase the specificity and utility of miRNAs as biomarkers for different diseases. Furthermore, combining miRNA profiles with other biomarkers or clinical information could enhance specificity.

Despite the challenges associated with using c-miRNAs as biomarkers for OA, their potential to aid in early diagnosis and inform treatment strategies is immense. As we unravel the complex interplay of miRNAs and other biological processes, the opportunities for their application in disease management and treatment only grow. Hopefully, with the evolution of our understanding and the development of robust analytical methods, c-miRNAs will serve as a linchpin in personalized medicine, thereby transforming how we diagnose, treat, and ultimately prevent OA.

## Figures and Tables

**Figure 1 ijms-24-13144-f001:**
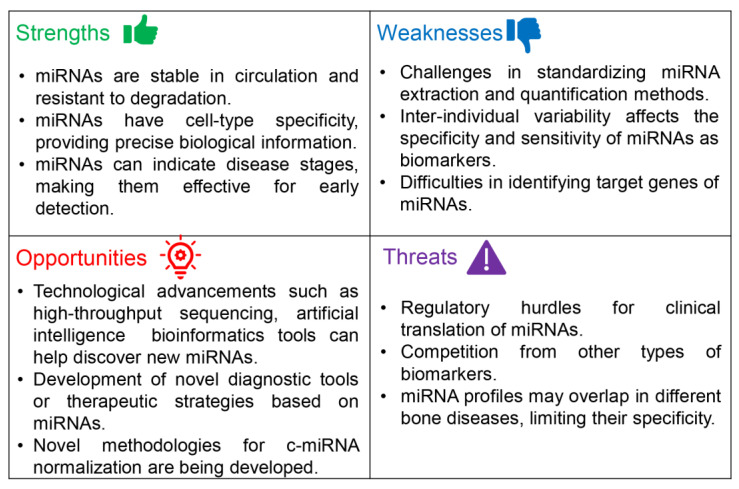
A SWOT analysis of the potential of c-miRNAs as biomarkers for the monitoring of the development and progress of OA.

**Figure 2 ijms-24-13144-f002:**
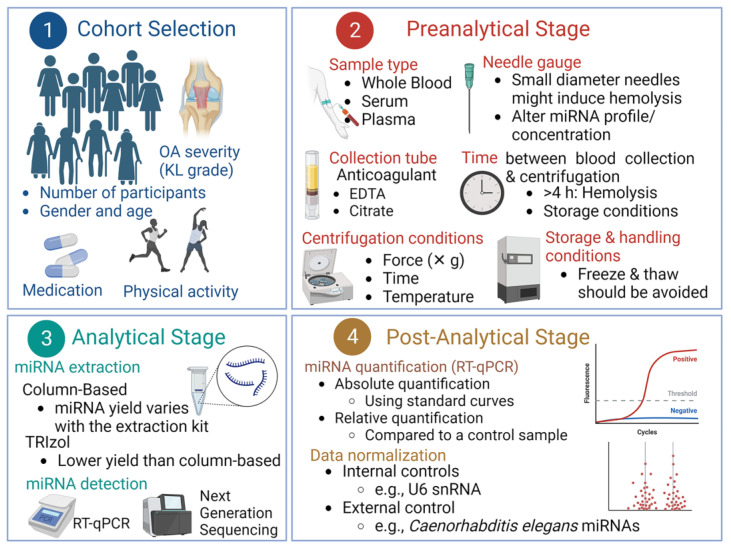
The transition of c-miRNAs as biomarkers for OA from basic research to clinical practice is hampered due to several obstacles and the lack of standardized protocols at four distinct levels: cohort selection, preanalytical, analytical, and post-analytical stages, as discussed in the text.

**Figure 3 ijms-24-13144-f003:**
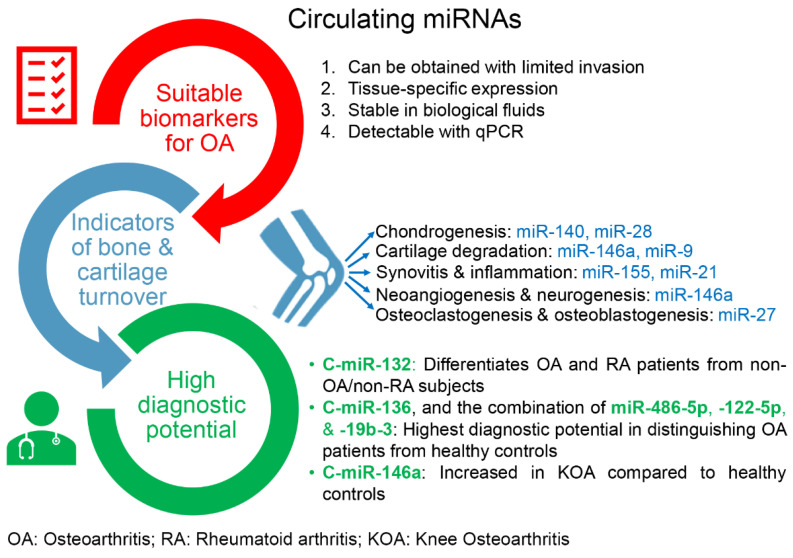
Circulating miRNAs (c-miRs) as promising prognostic and diagnostic biomarkers for osteoarthritis (OA). They are considered promising because they (i) fulfill the criteria of suitable biomarkers, (ii) can serve as indicators of bone and cartilage turnover, and (iii) exhibit high diagnostic potential. The miRNAs shown in this figure are indicative examples; their roles in the development of OA and potential as biomarkers for OA are extensively discussed in the text.

**Table 1 ijms-24-13144-t001:** Examples of miRNAs implicated in the development of osteoarthritis.

miRNA	Contribution in OA	Studies on the Role of miRNA in OA ^a^			
		Source/ Extraction	Detection/ Normalization	Main Findings/Regulation	Ref
9	Involved in chondrocyte hypertrophy and matrix degradation	Bone and cartilage/ TRIzol	RT-qPCR/ 18S RNA	Upregulated in OA bone and cartilage tissue	[116]
	Involved in the development of KOA through the NF-kB1 pathway in chondrocytes	KOA ^b^ cartilage/ TRIzol	RT-qPCR/ U6 snRNA	Downregulated in KOA compared to the control group	[117]
16	Involved in cartilage homeostasis and structure	**Plasma**/ miRNeasy kit	RT-qPCR/ MammU6s	Upregulated in KOA patients	[118]
		Articular cartilage from hip or KOA/N.A. ^c^	Northern Blot & RT-qPCR/U6 snRNA	Upregulated in chondrocytes of OA (KL ^d^ grades 3 and 4)	[119]
		**SF ^e^ and plasma**/ Phenol chloroform and High Pure miRNA Isolation Kit	RT-qPCR/ cel-miR-39	Downregulated in KOA patients	[120]
16-5p	Controls the development of osteoarthritis by targeting SMAD3 in chondrocytes	Cartilage/ TRIzol	RT-qPCR/ U6 snRNA	Upregulated	[121]
21	Influences apoptosis and autophagy of OA chondrocytes Suppresses chondrogenesis by directly targeting growth differentiation factor 5 Contributes to catabolic NF-κB signaling and MMP activation in response to inflammatory cytokines	Articular chondrocytes/ TRIzol	RT-qPCR/ U6 snRNA	Upregulated in KOA patients	[122]
27	Associated with chondrocyte degradation	**SF and serum** (animal model)/miRNeasy Mini Kit	RT-qPCR/ cel-miR-39	Upregulated in both serum and synovial fluid	[123]
27a-3p 27b-3p 27a-5p		**SF**/miRCURY RNA isolation kit	RT-qPCR/ mean Cq values of all detected miRNAs detected	Increased levels of miR27a-3p and 27b-3p and decreased levels of miR27a-5p in late-stage OA (compared to early-stage OA)	[124]
29a	miR-29 family regulates collagen synthesis and cartilage formation Involved in cartilage damage	Articular cartilage from hip or knee OA/N.A.	Northern blot & RT-qPCR/ U6 snRNA	Downregulated	[119]
29c		**Plasma**/ miRNeasy kit	RT-qPCR/ MammU6s	Upregulated in the plasma of KOA patients compared to healthy controls	[118]
30a	Members of the miR-30 family: Participate in bone development Mediate cartilage degradation	Cartilage/ TRIzol	RT-qPCR/ U6 snRNA	Upregulated in primary AC ^f^ cells from KOA patients compared to healthy controls	[125]
30b		**Plasma**/ miRNeasy kit	RT-qPCR/ MammU6s	Increased in the plasma of KOA patients compared to healthy controls	[118]
34a and 34b	Induce chondrocyte apoptosis and ECM degradation Contribute to the breakdown of cartilage and globally suppress signals related to tissue repair and remodeling	Bone and cartilage/ TRIzol	RT-qPCR/ 18S RNA	Both are overexpressed in the AC of KOA patients	[116]
126	Participates in anabolic and anticatabolic processes Regulates angiogenesis and de novo vascularization	**Plasma**/ miRNeasy kit	RT-qPCR/ MammU6s	Upregulated in the plasma of KOA patients compared to healthy controls	[118]
132	Downregulation of miR-132: Decreases cell proliferation and induces apoptosis in chondrocytes Promotes the expression of Bax protein and activated caspase-3/9 (inflammation)	**SF** and **Plasma**/ Phenol & High Pure miRNA Isolation Kit (Roche)	RT-qPCR/ cel-miR-39	Downregulated: Plasma miR-132 differentiated HCs ^g^ from patients with RA or OA	[120]
		**Serum**/TRIzol	RT-qRCT/ U6 snRNA	Downregulated	[126]
140	Influences chondrocyte differentiation and cartilage homeostasis and suppresses catabolic gene expression	**SF** and cartilage/microRNA Kit (Norgen)	RT-qRCT/ U6 snRNA	Downregulated: Expression levels correlated with OA severity	[127]
146a	Regulates inflammatory and catabolic gene expression	**Plasma**/ miRNeasy kit	RT-qPCR/ MammU6s	Upregulated	[118]
146a-5p	Regulates the expression of inflammatory cytokines	**Serum and cartilage**/ miRCURY RNA Isolation Kit	RT-qRCR/hsa-miR-103a-3p, -423-5p, and -191- 5p	Upregulated	[87]
		**Serum**/ miRCURY Kit	NGS ^g^/N.A.	Upregulated	[128]
186	Overexpression of miR-186 inhibits chondrocyte apoptosis in OA (see Ref. [129])	**Plasma**/ miRNeasy kit	RT-qPCR/ MammU6s	Upregulated	[118]
186-5p	Regulates chondrocyte apoptosis	**Serum**/ miRCURY Kit	NGS ^h^/ N.A.	Upregulated and significantly associated with incident KOA in women	[128]

^a^ OA: osteoarthritis; ^b^ KOA: knee osteoarthritis; ^c^ N.A.: not available; ^d^: KL: Kellgren–Lawrence score; ^e^ SF: synovial fluid, ^f^ AC: articular cartilage; ^g^ HCs: healthy controls; ^h^ NGS: next-generation sequencing. Biological fluids are indicated in bold.

**Table 2 ijms-24-13144-t002:** Examples of discrepancies among studies aiming to evaluate c-miRNAs as biomarkers for OA.

miRNAs	OA ^1^ Type	Cohort	Sample/ Collection Tube/ Centrifugation (×g/Time/Temp)	Extraction/ Quantification/ Normalization	Ref
146-5p	HOA ^2^	KL ^3^ grade I and II: n = 28 Control group: n = 2	Serum Collection tube ^4^: N.A. ^5^ 2000 g/10 min/N.A.	miRCURY RNA Kit/ RT-qRCR/ hsa-miR-103a-3p, -423-5p, & -191-5p	[87]
380 miRNAs	KOA ^6^	KL 2 or 3: n = 27 Control group: n = 27	Plasma ETDA Tube 1800 g/10 min/RT	miRNeasy kit/ RT-qPCR/ MammU6s	[118]
132	N.A	Cohort of 16 men: 8 OA patients 8 healthy controls	Serum/ Collection tube: N.A/ 2000 g/10 min/4 °C	TRIzol/ RT-qRCT/ U6 snRNA	[126]
140	KOA	KL 1–2: n = 10 KL 3: n = 10 KL 4: n = 10 Control group: n = 17	SF ^7^ & cartilage The collection and centrifugation conditions are not described	microRNA Kit (Norgen)/ RT-qPCR/ U6 snRN	[127]
19 miRNAs (Validation)	KOA	PM women: Screening: KL 2–3 = 10, control group: n = 10 Validation: KL 2–3 = 43, control group: n = 42	Serum The collection and centrifugation conditions are not described	miRCURY Kit/ NGS/ N.A	[128]
136	ΚOA	KL 2: n = 22 KL 3: n = 29 KL 4: n = 23 Control group n = 29	Plasma Sodium Citrate Centrifugation conditions are not available	RNAVzol LS/ RT-qPCR/ U6 snRNA	[142]
2578 miRNAs (Validation)	KOA	KOA who received celecoxib treatment for six weeks Screening KL 2: n = 4 & KL 3: n = 2 Validation KL 2: n = 159 & KL 3: n = 59	Plasma The collection and centrifugation conditions are not described	TRIzol/ RT-qPCR U6 snRNA	[148]

^1^ OA: osteoarthritis; ^2^ HOA: hip osteoarthritis; ^3^ KL: Kellgren–Lawrence score; ^4^ only one type of tube is commonly used for the collection of serum, i.e., without anticoagulant; ^5^ N.A; not available; ^6^ KOA: knee osteoarthritis; ^7^ SF: synovial fluid.

## Data Availability

No new data were created in this work.

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
