# Peer review of "Exploring the Feasibility of Circulating miRNAs as Diagnostic and Prognostic Biomarkers in Osteoarthritis: Challenges and Opportunities"

_ijms, 2023, doi:10.3390/ijms241713144_

Round 1

Reviewer 1 Report

The article "MicroRNAs: Basic Concepts" provides an overview of miRNA biogenesis, their biological roles in gene regulation, and their potential as circulating biomarkers. I have the following suggestions for you to improve.

1.         The article makes several statements, such as the origin of miRNAs in the nucleus, the processing steps involving Drosha-DGCR8 complex and Dicer, and the integration of one strand into the RISC complex. However, these claims lack proper citations to support their validity. It is crucial to provide credible references to established research in the field.

1.    The article mentions that miRNAs target nearly 30% of protein-coding genes and have significant contributions to various developmental processes and pathologies. However, it does not elaborate on specific examples or provide references to studies supporting these claims. A more comprehensive analysis of miRNA target genes and their roles in specific diseases would enhance the article's credibility.

2.       The article mentions that c-miRNAs are primarily found within exosomes, accounting for a large proportion of the total miRNA population detected in circulation. However, it fails to explain how exosomes protect and transport miRNAs in the extracellular environment. Adding more details on exosome functionality and the biological significance of c-miRNAs within exosomes would enhance the review's value. (similar or citation for your reference: ‘’Extracellular vesicles derived from host and gut microbiota as promising nanocarriers for targeted therapy in osteoporosis and osteoarthritis’’).

3.             The article mentions several miRNAs (e.g., miR-146a, miR-21, miR-140) and their potential roles in regulating inflammation, apoptosis, autophagy, and chondrogenesis. However, more details are needed to understand the specific mRNA targets of these miRNAs and how they contribute to the mentioned processes.

4.             Investigating the downstream effectors regulated by miRNAs, such as mesenchymal stem cells and transforming growth factor-β, and their impact on OA pathogenesis would provide valuable mechanistic insights.     

6.         to establish a miRNA as a robust biomarker, it is crucial to evaluate its diagnostic and prognostic value in large and diverse patient cohorts. Are there any studies that have independently validated the identified miRNA biomarkers, and do they consistently show the same trends across different populations?

7.         The article mentions different studies reporting the levels of specific miRNAs in plasma or serum samples from OA patients. It would be helpful to include a comparative analysis of the sensitivity, specificity, and diagnostic accuracy of these miRNAs in distinguishing OA patients from healthy controls. This analysis could provide a clearer picture of the most promising miRNA biomarkers for OA.

8.         to fully assess the potential of miRNAs as clinical biomarkers, it would be beneficial to discuss the challenges and prospects of translating these findings into routine clinical practice. What are the current hurdles in implementing miRNA-based diagnostics for OA, and what steps are being taken to overcome them?

9.         The article mentions the importance of selecting the appropriate blood fraction (whole blood, plasma, or serum) for c-miRNA analysis. However, it does not provide a detailed discussion on the advantages and limitations of each blood fraction. Could you elaborate on the reasons for selecting a specific blood fraction in different studies and the potential impact of using different blood fractions on the interpretation and comparison of c-miRNA results?

10.       The preanalytical stage plays a crucial role in ensuring the reliability and reproducibility of c-miRNA measurements. It would be beneficial to discuss the efforts made by researchers to standardize sample collection, processing, and storage conditions to minimize the impact of preanalytical factors on c-miRNA profiles. Are there any standardized protocols or guidelines that researchers should follow to ensure consistency in c-miRNA analysis for OA biomarker studies?

11. The article briefly mentions that certain drugs, such as heparin, aspirin, and anti-platelet therapies, may affect c-miRNA profiles. Could you provide more insights into the specific drugs and their potential impact on c-miRNA measurements?

Reviewer 2 Report

The review by Felekkis et al. explores the feasibility of circulating miRNAs as diagnostic and prognostic biomarkers in OA. Overall, the review provides a good overview on the same. However, I have some major concerns.

1. The introduction section needs to be shortened. Also, information in the introduction section overlooks some of the important aspects such as how miRNAs would serve as better biomarkers than the genetic and biochemical markers. This information needs to be added to the section.

2. The review is missing a section reviewing the role of miRNAs in bone development and their significance in OA.

3. Fig 1 is not required as the same information is provided in the text.

4. Information in Table 2 is not clear. What do the authors mean by 'discrepancies'? Do they mean different studies showing contradictory data for the same miRNA? If that is the case, then the authors need to provide two columns in the table, one citing studies reporting 'UP' and other citing studies reporting 'DOWN'-regulation of the same miRNA. In addition to this, the table looks incomplete as it does not mention all the studies cited in Section 3.

5. The review also lacks a scheme figure for the conclusion part showing the key miRNAs, their function, and their role in the development of OA.

Reviewer 3 Report

The authors' review "Exploring the Feasibility of Circulating miRNAs as Diagnostic and Prognostic Biomarkers in Osteoarthritis: Challenges and Opportunities" is a well written and comprehensive review of the subject. It enhances the readers understanding of the pathogenesis of osteoarthritis and provides valid arguments as to why the measurement of miRNAs may be useful in guiding the diagnosis and treatment of this most prevalent disease.

Author Response

We want to thank the Reviewer for the positive and encouraging feedback on our manuscript. We are pleased to hear that you found our review well-written and comprehensive. It was our aim to enhance the reader's understanding of the complex pathogenesis of osteoarthritis and shed light on the promising field of miRNA measurements in its diagnosis and treatment.

Round 2

Reviewer 1 Report

i have no further comments. accepted

Reviewer 2 Report

The review has been revised as suggested